# A Monocular-Visual SLAM System with Semantic and Optical-Flow Fusion for Indoor Dynamic Environments

**DOI:** 10.3390/mi13112006

**Published:** 2022-11-17

**Authors:** Weifeng Chen, Guangtao Shang, Kai Hu, Chengjun Zhou, Xiyang Wang, Guisheng Fang, Aihong Ji

**Affiliations:** 1College of Mechanical and Electronic Engineering, Quanzhou University of Information Engineering, Quanzhou 362000, China; 2College of Automation, Nanjing University of Information Science & Technology, Nanjing 210044, China; 3College of Mechanical and Automotive Engineering, Zhejiang University of Water Resources & Electric Power, Hangzhou 310018, China; 4Lab of Locomotion Bioinspiration and Intelligent Robots, College of Mechanical and Electrical Engineering, Nanjing University of Aeronautics & Astronautics, Nanjing 210016, China

**Keywords:** SLAM, dynamic, Mask R-CNN, optical flow, ORB-SLAM2

## Abstract

A static environment is a prerequisite for the stable operation of most visual SLAM systems, which limits the practical use of most existing systems. The robustness and accuracy of visual SLAM systems in dynamic environments still face many complex challenges. Only relying on semantic information or geometric methods cannot filter out dynamic feature points well. Considering the problem of dynamic objects easily interfering with the localization accuracy of SLAM systems, this paper proposes a new monocular SLAM algorithm for use in dynamic environments. This improved algorithm combines semantic information and geometric methods to filter out dynamic feature points. Firstly, an adjusted Mask R-CNN removes prior highly dynamic objects. The remaining feature-point pairs are matched via the optical-flow method and a fundamental matrix is calculated using those matched feature-point pairs. Then, the environment’s actual dynamic feature points are filtered out using the polar geometric constraint. The improved system can effectively filter out the feature points of dynamic targets. Finally, our experimental results on the TUM RGB-D and Bonn RGB-D Dynamic datasets showed that the proposed method could improve the pose estimation accuracy of a SLAM system in a dynamic environment, especially in the case of high indoor dynamics. The performance effect was better than that of the existing ORB-SLAM2. It also had a higher running speed than DynaSLAM, which is a similar dynamic visual SLAM algorithm.

## 1. Introduction

Simultaneous localization and mapping (SLAM) technology is a prerequisite for many mobile robot applications. It can simultaneously complete a robot’s localization and construct a map of its surrounding environment without needing any prior environmental information [1]. With the rapid development of mobile robots, autonomous driving, virtual reality, and drones, SLAM technology has attracted more and more attention. SLAM technology can be divided into laser SLAM systems and visual SLAM systems, according to the different sensors used. With the development of computer vision and deep learning and the improvements in hardware computing abilities, research on vision-based SLAM systems has been continuously and widely applied in autonomous driving, mobile robots, and drones, among other fields [2]. Vision-based SLAM systems use cameras as their primary sensors. Due to their low costs and easy installation, these cameras can obtain richer scene information and facilitate combination with other sensors. Vision-based SLAM systems have been favored by many scholars [3]. Monocular-visual SLAM systems have become the first choice for many researchers due to their low costs, small sizes, and convenience.

Nowadays, vision-based SLAM technology has matured and many excellent works have emerged, for example, MonoSLAM [4], ORB-SLAM2 [5], Vins-Mono [6], etc., which have made remarkable achievements in many fields. Such visual SLAM algorithms are mainly based on the static-environment hypothesis. As a result, most of the existing visual SLAM systems are based on the static-environment hypothesis. In static environments, the use effect is good if the dynamic objects in a scene make the system positioning precision suffer from more significant deviations and severe impact effects, even resulting in system failure [7]. Under this assumption, SLAM application scenarios must be static or have a low dynamic range and any changes in the field of view must only come from the camera’s motion [8]. However, in reality, SLAM application scenarios do not often meet this static-environment assumption. There are usually many moving objects in practical application scenarios, so the application scope of traditional vision-based SLAM systems is minimal. For example, in indoor environments, moving people are ubiquitous dynamic objects. When people enter the camera’s field of view, most of the camera’s field of view can become blocked. This situation may lead to fewer static features that can be extracted by the SLAM system or the failure to extract any stable stationary feature points, thereby leading to the direct interruption of the system [9]. A feasible method to counter this problem is to use different sensors and fuse their collected information to address the interference of dynamic objects in SLAM systems. For example, a camera combined with an IMU can effectively deal with some dynamic scenes. However, the fusion of multiple sensors requires a high computational load, which is not economical [10]. Usually, their low costs, easy calibration, and portability make monocular cameras the preferred sensors for visual SLAM systems.

Similar studies have been detailed in our previous work [11]. Therefore, in this study, we mainly focused on how to solve the dynamic SLAM system problem using monocular cameras. To solve the dynamic visual SLAM-system problem, we extended the work of ORB-SLAM2. A novel monocular dynamic visual SLAM algorithm was proposed to improve the robustness of a visual SLAM system in dynamic scenarios. In this study, the system’s input was an RGB image obtained by a monocular camera, which could be applied to almost all robot platforms. We validated the proposed algorithm using public datasets and real scenarios. Our experimental results showed that this method could effectively improve the system-performance degradation caused by dynamic-object occlusion in cameras. The algorithm could also be used for different tasks, such as the autonomous navigation, positioning, and exploration of robots.

The main contributions of this paper are as follows:(1)A novel monocular visual SLAM system based on ORB-SLAM2 which can achieve more accurate localization and mapping in dynamic scenes;(2)An improved Mask R-CNN which can more accurately segment prior highly dynamic objects in indoor dynamic environments to meet the requirements of the SLAM algorithm in dynamic scenes;(3)The combination of the optical-flow method and the geometric approach, which offers a suitable and efficient dynamic-object processing strategy to remove highly dynamic objects in dynamic scenes.

The rest of this paper is structured as follows. Section 2 discusses related works, including classical visual SLAM algorithms and dynamic visual SLAM systems. Section 3 describes our proposed system in detail. Section 4 details the experimental process and compares, analyzes, and discusses the experimental results. Finally, the work in this paper is discussed and summarized in Section 5.

## 2. Related Works

In dynamic scenes, the performance of current visual SLAM systems decreases significantly due to the interference of dynamic objects [12]. Recently, many researchers have focused on SLAM systems for use in dynamic environments. The main challenge in solving this problem is effectively detecting and filtering out dynamic features. The tracking process should avoid using components that are extracted from moving objects as much as possible [13]. This section reviews some of the best representative dynamic SLAM algorithms. In addition, we explain the related techniques for dynamic visual SLAM systems in Section 2.1 and Section 2.2.

### 2.1. Visual SLAM Systems

Traditional vision-based SLAM research has yielded many outstanding achievements. In 2007, Davidson et al. [4] first proposed MonoSLAM, the first SLAM system to achieve monocular real-time capture. Subsequently, the PTAM presented by Klein et al. [14] creatively divided systems into two threads: tracing and mapping. This became the benchmark for subsequent SLAM researchers. In 2014, the LSD-SLAM algorithm was proposed by Engel et al. [15], which clarified the relationships between pixel gradients and direct methods. Meanwhile, Forster et al. [16] suggested a fast semi-direct monocular visual odometer (SVO), which combined feature points and the direct-tracking optical-flow method, which saved computational resources in terms of tracking and matching but had defects in terms of feature extraction.

Using feature points is crucial to building complete and robust SLAM frameworks. On the one hand, feature extraction and matching can ensure pose estimation accuracy in SLAM tracking [17]. On the other hand, the feature-point method can extract more useful information from visual images, such as semantics, object recognition, feature localization, etc. [18]. In 2015, the ORB-SLAM algorithm was proposed by Mur-Artal et al. [19], which innovatively used three tracking threads, local mapping, and closed-loop detection simultaneously, effectively reducing cumulative errors. The closed-loop detection thread used the bag of words (DBoW) model for closed-loop detection and correction, which achieved good results in terms of processing speed and map construction accuracy. In subsequent years, the team also launched ORB-SLAM2 and ORB-SLAM3 [20]. The ORB-SLAM algorithms are some of the most widely used visual SLAM schemes due to their ability to work in real time in CPUs and their good robustness. They are also some of the best feature-point extraction methods [21].

ORB-SLAM2 is one of the best and most stable visual SLAM algorithms. It is a complete SLAM system for monocular, stereo, and RGB-D cameras. It tracks ORB features as map points throughout the frames. The calculation and matching of ORB features are fast and have good invariance to viewpoints [22]. However, this algorithm still leaves much to be desired in terms of dealing with dynamic environmental issues, so further improvements are necessary. Many classical monocular visual SLAM systems have been developed over recent decades, but most fail when dynamic scenes dominate. Therefore, how to solve the problem of the robustness of dynamic-scene monocular-visual SLAM algorithms has become a hot topic for many researchers.

### 2.2. SLAM Systems in Dynamic Scenarios

In general, the visual-SLAM-system problem can be represented by the following observation model:(1)$$z_{k,j}=h\left(x_{k},y_{j},v_{k,j}\right)$$
where $x_{k}$ represents the camera’s position in the world coordinate system at time *k*; $y_{j}$ represents the position of the *j* landmark point; $z_{k,j}$ is the pixel coordinate in the image that corresponds to the landmark observed by the camera at time *k*; *h* is a nonlinear equation; and $v_{k,j}∼N\left(0,Q_{k,j}\right)$ is the Gaussian equation (assuming zero mean variance), which is the covariance matrix of the observation equation. Based on this, the error term can be defined as follows:(2)$$e_{k,j}=z_{k,j}-h\left(x_{k},y_{j}\right)\;$$

Then, the cost function can be described as follows:(3)$$J\left(x\right)=\frac{1}{2}\sum_{k}^{m}\sum_{j}^{n}e_{k,j}{\left(x\right)}^{T}Q_{k,j}^{-1}\left(x\right)\;$$

This is a typical nonlinear slightest squares problem [23], where *x* represents the camera pose and landmark. The correct solution to this problem can be obtained in static environments by iteration. However, moving landmarks can lead to mismatched observations in dynamic environments, resulting in incorrect data-association problems. As shown in Figure 1, dynamic objects cause cameras to fail to capture data accurately; thus, how to reduce the impact of dynamic objects on SLAM systems has become the goal of many researchers.

Traditional visual SLAM algorithms assume that objects in the environment are static or have a low range of motion, which affects the applicability of visual SLAM systems in actual scenes [24]. When there are dynamic objects in the background, such as people walking back and forth, vehicles traveling on a road, and moving pets, they introduce incorrect observation data into the system and reduce the accuracy and robustness of the system. Traditional methods reduce the influence of outliers on the strategy using the RANSAC algorithm. However, if dynamic objects occupy most of the image area or they are fast, then reliable observation data still cannot be obtained [25].

To solve the SLAM-system problem in dynamic environments, some scholars have used the optical-flow method to filter out moving objects in a scene. These algorithms are sensitive to environmental changes, such as illumination, and have poor robustness. The development of deep learning has brought about new solutions to this problem [26]. Many excellent semantic-segmentation algorithms have been proposed, such as SegNet [27], DeepLab [28], and Mask R-CNN [29], etc. These semantic-segmentation algorithms can realize the pixel-level classification of an image to obtain the boundaries and semantic information of the objects in the image. However, these semantic methods cannot distinguish whether dynamic objects are currently in motion or at rest, so they can only detect potential dynamic objects. Some objects, such as people, cars, and animals, are usually in action, so removing them does not cause adverse consequences [30]. However, in scenes that lack texture, such as parking lots, if all potential dynamic objects are considered to be moving objects and are removed, then systems may not obtain enough reference objects, resulting in inaccurate positioning or even lost positioning. In the scenario of a parking lot, parked vehicles (potential dynamic objects at rest) are good temporary references that can be used to optimize positioning, but to use temporary reference objects effectively, it is necessary to detect the motion state of potential dynamic objects [31].

With the help of a semantic-segmentation networks, dynamic objects in an environment can be eliminated effectively. Several excellent robust SLAM systems for use in dynamic scenarios and deep-learning algorithms have been proposed which can significantly improve performance. Zhong et al. [32] used an SSD to detect objects in an image, filtered out all feature points on objects that were a-priori marked as dynamic, and then performed subsequent camera pose estimation. Bescos et al. [33] first combined a Mask R-CNN [34] with multi-view geometry to judge whether dynamic segmentation results moved. Yu et al. [35] applied the SegNet network to perform the pixel-level semantic segmentation of images. They combined the network with a motion consistency strategy to filter out the dynamic objects in a scene. In addition, this network also generated dense semantic octree graphs, which helped to further deepen the understanding and mapping of dynamic scenes. Cui et al. [36] integrated optical flow into the RGB-D mode of ORB-SLAM2, based on semantic information and their proposed semantic optical-flow SLAM system, which was a visual-semantic SLAM system for use in dynamic environments. The semantic optical-flow method is a tightly coupled method that can fully use the dynamic characteristics of features hidden in semantic and geometric information and remove dynamic features more effectively and reasonably.

Deep-learning-based methods mainly use semantic information to detect and segment objects and remove outliers. In the real world, moving objects are not always in motion. For example, a moving car may suddenly stop at the side of the road, changing from a dynamic state to a stationary state. Since moving objects in the real world are usually people, animals, vehicles, and other semantic objects, they can be considered to be potential dynamic objects. Since latent dynamic objects are determined by their semantics, the semantic information of an object and the pixels in an image can be obtained using object-detection and semantic-segmentation technology. Based on semantics, whether an object in an image is a latent dynamic object or not can be determined. Inspired by the above works, we proposed an efficient and reliable monocular-visual SLAM method for filtering out dynamic feature points from natural scenes. Our experimental results showed that the proposed method could significantly improve positioning accuracy and real-time performance in dynamic environments.

## 3. System Overview and Approach

In this section, we detail the proposed SLAM algorithm for use in dynamic environments. This section describes our improved approach from the following three main perspectives. Firstly, we introduce the overall framework of our improved dynamic VSLAM system, which was based on ORB-SLAM2. It mainly consisted of four core threads: dynamic feature removal, trace, local mapping, and closed-loop detection. Secondly, we detail the semantic-segmentation approach that we adopted. Finally, our method for screening dynamic feature points is introduced and analyzed. This method assumed a convenient and efficient dynamic-object screening and removal strategy, which effectively improved the system’s operational efficiency. Specifically, Section 3.1 describes the overall framework of the algorithm, Section 3.2 introduces the semantic-segmentation method adopted and Section 3.3 presents the proposed efficient processing strategy and removal method for dynamic objects.

### 3.1. System Overview

The dynamic SLAM algorithm proposed in this paper was an improved version of ORB-SLAM2. As one of the most complete, stable, and widely used open-source visual SLAM algorithms, ORB-SLAM2 has been favored by many researchers. Our visual odometer was developed based on the original framework of ORB-SLAM2. Based on the actual tracking, local construction, and loop detection threads, a moving-object processing thread was added to remove the impact of dynamic objects on the system. A diagram of the improved system framework is shown in Figure 2. The local map construction and loop detection threads were the same as those in ORB-SLAM2. The former was used to maintain a map of the surrounding environment and perform bundle adjustment (BA) to optimize the camera pose. The latter was used for loop detection and fusion to eliminate accumulated errors.

We also added a dynamic-object processing thread to the front of the tracking thread. The dynamic-object processing included three steps: the semantic segmentation of prior highly dynamic objects, optical-flow feature tracking, and dynamic-feature elimination. The process was as follows: (1) RGB images were input into the segmentation network, the prior highly dynamic objects in the image were segmented and their feature points were removed, and the ORB features from the RGB images were extracted simultaneously; (2) a motion continuity check was performed to extract the dynamic points, which could be divided into two sub-steps (fundamental matrix estimation and polar constraint) or, more specifically, into three steps (the optical-flow tracking estimation of the fundamental matrix between two adjacent frames, the extraction of dynamic points by the polar constraint, and the establishment of a set of dynamic feature points); (3) after the motion continuity check was completed, the dynamic-feature removal module used the dynamic feature points to determine whether the objects in the image were moving, then the ORB feature points of the moving object were discarded, and the remaining static feature points were input into the SLAM system for subsequent tracking and map construction.

The semantic-segmentation thread used an improved Mask R-CNN to segment the prior highly dynamic objects in the scene. Then, a motion-consistency detection algorithm based on optical flow and geometric constraints was used to further detect the potential dynamic feature-point outliers. Then, the feature points of the dynamic objects were eliminated and the relatively stable static feature points were used to perform pose estimation. Our method only used filtered stationary feature points for tracking, which significantly improved the system’s operating efficiency and could improve the system’s robustness in dynamic environments.

### 3.2. Semantic Segmentation

We used an improved Mask R-CNN for the semantic segmentation of prior highly dynamic objects in images. The Mask R-CNN extracted features using ResNet [37] and formed a backbone network via a feature pyramid network (FPN) [38]. The generated feature map shared the convolution layer and the feature pyramid performed different combinations of the extracted stage features. On the one hand, a region proposal network (RPN) [39] was used to determine the foreground and background for binary classification and generate candidate boxes. On the other hand, the generated candidate box corresponded to a pixel in the feature map, which was matched using the RoI Align [40] operation [41]. After that, one branch was used for classification and regression, while the other was used for segmentation to generate masks. We focused on indoor dynamic environments, so “people” were our main focus. In this study, the Mask R-CNN was fine-tuned and its process is shown in Figure 3. A cross-layer feature pyramid network (CFPN) module and branch were added for dynamic environments. The purpose of this was to use contextual semantic information more fully by redesigning the feature-extraction backbone network. Unlike the original network, the feature group processed by the FPN generated candidate boxes via an RPN. It was only used for classification and regression after the RoI Align operation, while the feature group and candidate boxes fused by the CFPN only entered the Mask R-CNN branch for the segmentation task after the RoI Align operation. Since the direct addition of multi-dimensional features was not conducive to the subsequent full-convolution segmentation task, an improved CFPN module was designed for the Mask R-CNN branches which could more fully utilize the semantic information from different layers to improve network performance. The features extracted by ResNet were denoted as C1–C5 and a “1 × 1” convolution kernel convolved each layer’s characteristics to maintain the feature map’s size and reduce the dimensions, simultaneously. After the C3–C5 convolutions, upsampling was performed to double the feature-map size and integrate the features from the lower layer.

Each fused feature was convolved with a “3 × 3” convolution kernel to eliminate the aliasing effect after upsampling and achieve dimensionality reduction. Then, “m5” was obtained by the direct upsampling and convolution of C5 and “m2” to “m4” were obtained by upsampling high-level features and fusing low-level features, respectively. The feature groups (m2, m3, m4, and m5) were used for Mask R-CNN branches. The CFPN fused the extracted stage features with the low-level features via upsampling to form m2, m3, m4, and m5. Combined with the candidate box, the CFPN used the RoI Align operation for the Mask R-CNN branches. Firstly, continuous convolution operations with a convolution-kernel size of “3 × 3” and 256 channels were carried out. After a “2 × 2” transposed convolution to double the feature area, the pixel segmentation results for the corresponding category in the candidate box were finally output, which was equivalent to the instance segmentation of people. The test results are shown in Figure 4.

Prior highly dynamic objects in an environment could be segmented using our Mask R-CNN. According to prior knowledge, we divided objects in an everyday environment into three types: highly dynamic, medium dynamic, and static. In everyday life, people and animals are the most common dynamic objects. People, cats, and dogs are defined as highly dynamic objects in this paper. As for cars, chairs, boxes, and other things that can change their motion state under human influence, they are defined as medium dynamic objects. In semantic segmentation, we only segmented highly dynamic objects. People are the most common objects in everyday indoor environments, so we only conducted the test for people, which significantly improved the system’s operational efficiency.

### 3.3. Dynamic-Object Removal

After filtering out the preliminary highly dynamic objects using the fine-tuned Mask R-CNN, the optical-flow method was used to match the remaining feature-point pairs. A fundamental matrix was calculated using the matched feature-point pairs. Finally, the environment’s fundamental dynamic feature points were filtered out using the polar geometric constraint. In this study, the improved system could effectively filter out the feature points of dynamic targets while ensuring the system’s stable operation. The following specific practices are shown in Figure 5:(1)The optical-flow method was used to calculate the matching points of two adjacent frames to form matching point pairs;(2)A fundamental matrix for the two adjacent frames was calculated using the matching point pairs;(3)The polar line of the current frame was calculated using the base matrix;(4)The distances between all matching points and the polar line were established and any matching points whose distances exceeded the preset threshold were classified as moving points.

#### 3.3.1. Optical-Flow Tracking

The optical-flow method is a common method in motion detection, in which the calculation of partial pixel motion in the whole image is called sparse optical flow [42]. Sparse optical flow is dominated by Lucas–Kanade optical flow, also known as LK optical flow [43]. After prior highly dynamic-object segmentation removal, we performed optical-flow tracking on the remaining feature points. To improve computational efficiency, only the optical-flow field of the ORB feature point after removing the prior highly dynamic object was calculated, so LK optical flow was used. The motion information of objects in images can be obtained by analyzing the optical-flow vectors of all pixels [44]. However, for sparse optical flow, the obtained optical-flow vectors are not dense enough to specifically analyze which pixel motion is the static background motion and which is the dynamic-object motion [45]. Therefore, our dynamic-object detection algorithm did not conduct a motion-consistency analysis for sparse optical flow and instead only used sparse optical flow to track some feature points. The corresponding relationships between feature points could be obtained directly during optical-flow tracking without employing the descriptor calculation process. The feature points traced between frames could solve the fundamental camera matrix. Then, the polar constraint was used to identify the dynamic objects, which could be recognized by the number of dynamic points of the potential moving objects. An optical-flow diagram is shown in Figure 6, where It1, It2, and It3 are the grayscale values of the image at times t1, t2, and t3, respectively.

An image’s grayscale level can be regarded as a function of time. At time *t*, the grayscale level of the ORB feature point located at (x,y) in the image could be written as I(x,y,t), whereas at time t+dt, it moves to (x+dx,y+dy). From the invariance in the grayscale level, we could obtain the following:(4)I(x,y,t)=I(x+dx,y+dy,t+dt)

Using Taylor’s expansion of the above equation and preserving the first-order term, we could obtain the following:(5)$$I\left(x+dx,y+dy,t+dt\right)\approx I\left(x,y,t\right)+\frac{\partial I}{\partial x}dx+\frac{\partial I}{\partial y}dy+\frac{\partial I}{\partial t}dt\;$$

Due to the assumption of a constant grayscale level, the grayscale values of feature points at time *t* and t+dt were equal and the following could be obtained:(6)$$\frac{\partial I}{\partial x}dx+\frac{\partial I}{\partial y}dy+\frac{\partial I}{\partial t}dt=0\;$$

By dividing both sides by dt, we could obtain the following:(7)$$\frac{\partial I}{\partial x}\frac{dx}{dt}+\frac{\partial I}{\partial y}\frac{dy}{dt}=-\frac{\partial I}{\partial t}\;$$
where the velocities of pixels on the *X* and *Y* axes are dx/dt and dy/dt (denoted as *u* and *v*), the direction gradients of the image on the *X* and *Y* axes are ∂I/∂x and ∂I/∂y (denoted as Ix and Iy), and ∂I/∂t represents the change in grayscale level over time (denoted as It). Therefore, the above equation could be transformed into the following matrix form:(8)$$\left[I_{x}I_{y}\right]\left[\begin{array}{c} u\\ v \end{array}\right]=-I_{t}\;$$

Equation (Equation 8) had two unknowns. To solve *u* and *v*, we assumed that the motion of pixels in a window was the same and set a “w·w” window with 2w pixels. Since the motion of pixels in this window was the same, we could obtain the following $w^{2}$ equations:(9)$${\left[I_{x}I_{y}\right]}_{k}\left[\begin{array}{c} u\\ v \end{array}\right]=-I_{tk}\left(k=1,2,...,w^{2}\right)\;$$

Remember the following two parameters:(10)$$A=\left[\begin{array}{c} {\left[I_{x}I_{y}\right]}_{1}\\ \vdots \\ {\left[I_{x}I_{y}\right]}_{k} \end{array}\right],b=\left[\begin{array}{c} I_{t1}\\ \vdots \\ I_{tk} \end{array}\right]\;$$

Then, Equation (Equation 9) could be written as the following formula:(11)$$A\left[\begin{array}{c} u\\ v \end{array}\right]=b\;$$

After using the least squares method, we could obtain the following:(12)$${\left[\begin{array}{c} u\\ v \end{array}\right]}^{*}=-{\left(A^{T}A\right)}^{-1}A^{T}b\;$$

After obtaining the motion velocity *u* and *v* of pixels, the position of a specific pixel could be estimated in several images, so the optical-flow method was used to track and match the ORB feature points. The optical-flow method can effectively avoid false matching. Finally, the solution with the best effect was obtained through several iterations to track the pixels. After the corresponding matching points were obtained, the fundamental matrix *F* of the matching point pairs was calculated. Finally, the polar geometry constrained all feature points and filtered out the scene’s remaining fundamental dynamic feature points.

#### 3.3.2. Geometric Constraints

After the fundamental matrix was obtained via LK optical-flow matching, the corresponding polar lines of the feature points were calculated using the positions of the fundamental matrix and feature points. When the distances between the feature points and the polar line were more significant than a specific value, they were classified as dynamic feature points.

Supposing that a camera observed the same point *P* in space from different angles. According to the pinhole camera model, the pixel coordinates of the point $x={\left[u\;v\;1\right]}^{T}$ on the two images, namely, $x_{1}$ and $x_{2}$, conform to the following constraints:(13)$$\left\{\begin{array}{c} s_{1}x_{1}=Kp\\ s_{2}x_{2}=K\left(Rp+T\right) \end{array}\right.\;$$
where *K* represents the internal reference matrix of the camera, *R* and *T* represent the rotation and translation matrices between the two camera coordinate systems, respectively, and *S* represents the depth information of the pixels. In the ideal case, the coordinates of the matching point pairs in the two images conform to the following constraint:(14)$${x_{2}}^{T}Fx_{1}=\left[u_{2}\;v_{2}\;1\right]F\left[\begin{array}{c} u_{1}\\ v_{1}\\ 1 \end{array}\right]=0\;$$
where *F* is the fundamental matrix. However, in actual scenes, because the photos collected by a camera are not ideal pictures, there is a certain degree of distortion and noise, so the points between adjacent frames cannot perfectly match the upper polar line *L*, as follows:(15)$$D=\frac{\left|{x_{2}}^{T}Fx_{1}\right|}{\sqrt{{∥X∥}^{2}+{∥Y∥}^{2}}}\;$$

If the distance *D* is greater than the threshold, it is considered that the point does not meet the polar constraint. These points move with an object’s motion, creating mismatches, which are dynamic points. Therefore, in this study, all feature points that did not conform to the polar constraint were filtered. As shown in Figure 7, our efficient dynamic-object processing strategy could accurately identify dynamic and static objects in highly dynamic environments.

## 4. Experimental Results

Experiments were carried out on public datasets to demonstrate the effectiveness of the proposed algorithm in more detail. The datasets we used were two TUM datasets and one Bonn RGB-D Dynamic dataset, which is a more realistic dataset with highly dynamic scenes. We only tested the above datasets monocularly. In addition, we also tested our algorithm on natural environments. The dataset experiments were conducted on a computer with an Intel i7 processor, a 16G CPU, and the Ubuntu 18.04 operating system. In this study, the accuracy of the ORB-SLAM2 algorithm was compared to that of the ORB-SLAM2 algorithm to analyze their robustness in dynamic environments. At the same time, we also compared similar dynamic visual SLAM algorithms. According to our results, the performance of the proposed system in dynamic indoor environments was better than that of other existing systems and our method was especially more efficient in highly dynamic scenes.

We used absolute pose error (APE) and relative pose error (RPE) to evaluate the positioning accuracy of the algorithms. The absolute trajectory error was used to assess the global consistency of estimated trajectories, indicating the differences between the estimated camera pose values and the actual values in each frame. The relative pose error was used to evaluate the local rotation of estimated trajectories or translation errors, indicating the differences between the estimated pose transformation matrices between two frames separated by fixed time differences and the real pose change matrices between two frames [46].

### 4.1. Indoor Accuracy Test on the TUM Dataset

In 2012, the TUM RGB-D dataset was published by the Computer Vision Group of the Technical University of Munich and it is currently one of the most widely used SLAM datasets [47]. This dataset comprises images of 39 different indoor scenes and the real motion trajectories of a camera, which were collected by a Microsoft Kinect sensor and an external motion capture system. It includes static, dynamic, and static–dynamic environment scenes, so it is widely used for the performance evaluation of visual SLAM algorithms [48]. In the TUM RGB-D dataset, the dynamic scenes mainly consist of two types: “walking” and “sitting”. In the “walking” series, two people walk around scenes and interact with surrounding objects. These scenes are considered as highly dynamic environments in this paper, which posed certain challenges for the performance and robustness of the SLAM system. We tested our method on the “rgbd_dataset_freiburg3_sitting_xyz” and “rgbd_dataset_freiburg3_walking_xyz” sequences and compared its performance to those of the ORB-SLAM2 and DynaSLAM algorithms.

In the “rgbd_dataset_freiburg3_sitting_xyz” sequence, which consists of two people who are sitting and chatting, there are only a few dynamic features and very little disturbance to the environment, so it is regarded as a low-dynamic environment in this paper. As shown in Figure 8, we compared the trajectory accuracies of ORB-SLAM2, DynaSLAM, and our approach. It can be seen that in low-dynamic environments, ORB-SLAM2 still had good robustness and could achieve good test results. However, our method could effectively improve the global trajectory accuracy, especially in the *Z*-axis direction, and had more advantages than the other two methods. As can be seen in the figure, the performance of the three algorithms in low-dynamic sequences was similar. From the perspective of RMSE, the three algorithms all performed well in terms of APE, although the proposed method had a slight advantage. In terms of RPE, the proposed algorithm showed obvious benefits over ORB-SLAM2 and DynaSLAM.

The “rgbd_dataset_freiburg3_walking_xyz” sequence, which is more challenging for SLAM systems because it contains people walking back and forth, is considered as a highly dynamic environment sequence in this paper. As shown in Figure 9, we compared the trajectory accuracies of the three algorithms. The experimental results proved that our method could effectively reduce the interference caused by dynamic-object environments and the system’s robustness in highly dynamic environments. We collected data such as root mean square error (RMSE), median, mean and standard deviation (StD), and took the most commonly used and most representative RMSE as the main evaluation index. We bolded the RMSE data from the three methods to make the comparison more intuitive for the reader. Specific data are shown in Table 1 and Table 2. Specifically, for the “sitting_xyz” data series, the proposed method improved the APE RMSE value by about 9.7% and the RPE RMSE value by about 39.1% compared to ORB-SLAM2.

For the “walking_xyz” dataset, the proposed method improved the APE RMSE value by about 8.3% compared to ORB-SLAM2; however, the proposed method failed to achieve better results than ORB-SLAM2 in terms of RPE RMSE. The experimental results from the TUM dataset showed that the proposed method could obtain higher global and local pose accuracy than the ORB-SLAM2 algorithm in low dynamic environments. In highly dynamic datasets, the proposed algorithm could still achieve lower absolute trajectory errors than ORB-SLAM2, thereby improving global pose accuracy in dynamic environments.

Table 3 shows the partial time indices of the three algorithms; we bolded the test data obtained by the algorithm in this paper to make it easier for readers to compare. According to the median and mean tracking times, ORB-SLAM2 was far more efficient than DynaSLAM and the proposed method. The experimental results showed that system speed was significantly reduced when the semantic-segmentation network was added; however, the proposed algorithm was faster than the DynaSLAM algorithm. Although it did not achieve the effect of real-time operation, our method could still significantly improve the running speed of existing dynamic SLAM systems.

### 4.2. Indoor Accuracy Test on the Bonn RGB-D Dynamic Dataset

The Bonn RGB-D Dynamic dataset contains highly dynamic sequences. The dataset comprises 24 dynamic sequences in which people are performing different tasks, such as carrying boxes or playing with balloons, and two static sequences. For each scene, the actual location of the sensor is provided, which was recorded using the Optitrack Prime 13 motion capture system [49]. The sequences are in the same format as those in the TUM RGB-D dataset, so the same evaluation tools could be used. In addition, ground-truth 3D point clouds are provided for the static environments, which were recorded using a Leica BLK360 ground-based laser scanner. The dataset contains a variety of highly dynamic scenes, which are more suitable for real life. Therefore, we selected the “rgbd_bonn_moving_nonfaceting_box2” data series for testing, which includes an everyday primary dynamic environment which could simulate indoor highly dynamic environments more realistically.

The ORB-SLAM2 and DynaSLAM algorithms were compared to the proposed algorithm and the results are shown in Figure 10. It can be seen that the performance of the ORB-SLAM2 algorithm, which was initially good in low-dynamic environments, degraded when using the Bonn highly dynamic dataset. For the “rgbd_bonn_moving_nonface-ting_box2” sequence, the testing effect was markedly decreased because DynaSLAM could not accurately judge the dynamic objects due to the influence of the boxes.

As shown in Table 4, we collected data such as root mean square error (RMSE), median, mean and standard deviation (StD). In addition, we took the most commonly used and most representative RMSE as the main evaluation index; in order to help readers more intuitively understand, we have bolded it in the table. In addition, it can be seen from Figure 11 that the proposed method could better improve the relative pose accuracy in highly dynamic environments. The data showed that our approach was more advantageous in reducing the relative pose error. The proposed method achieved a small number of errors on this sequence. However, in terms of improving the absolute pose accuracy, the method proposed in this paper does not show any more obvious advantages than the ORB-SLAM2. By analyzing the data and picture sequence in Table 4, it can be inferred that, in the “rgbd_bonn_moving_nonfaceting_box2” sequence, because the dynamic object did not appear in the camera field of view for a long time, it did not have a great impact on the global pose, so the feature extraction performance of ORB-SLAM2 was still good, so there is little difference in the APE obtained by the three methods. However, concerning the aspect of local pose estimation, at some point, because there are only people and a box in the camera field of vision, the box motion with rich feature points will cause serious interference to the system. In this case, because there are many feature points on the dynamic box, ORB-SLAM2 cannot handle the ground feature points on the dynamic box well. Our method can achieve higher relative trajectory accuracy by eliminating the influence of the situation.

**Figure 9 micromachines-13-02006-f009:**
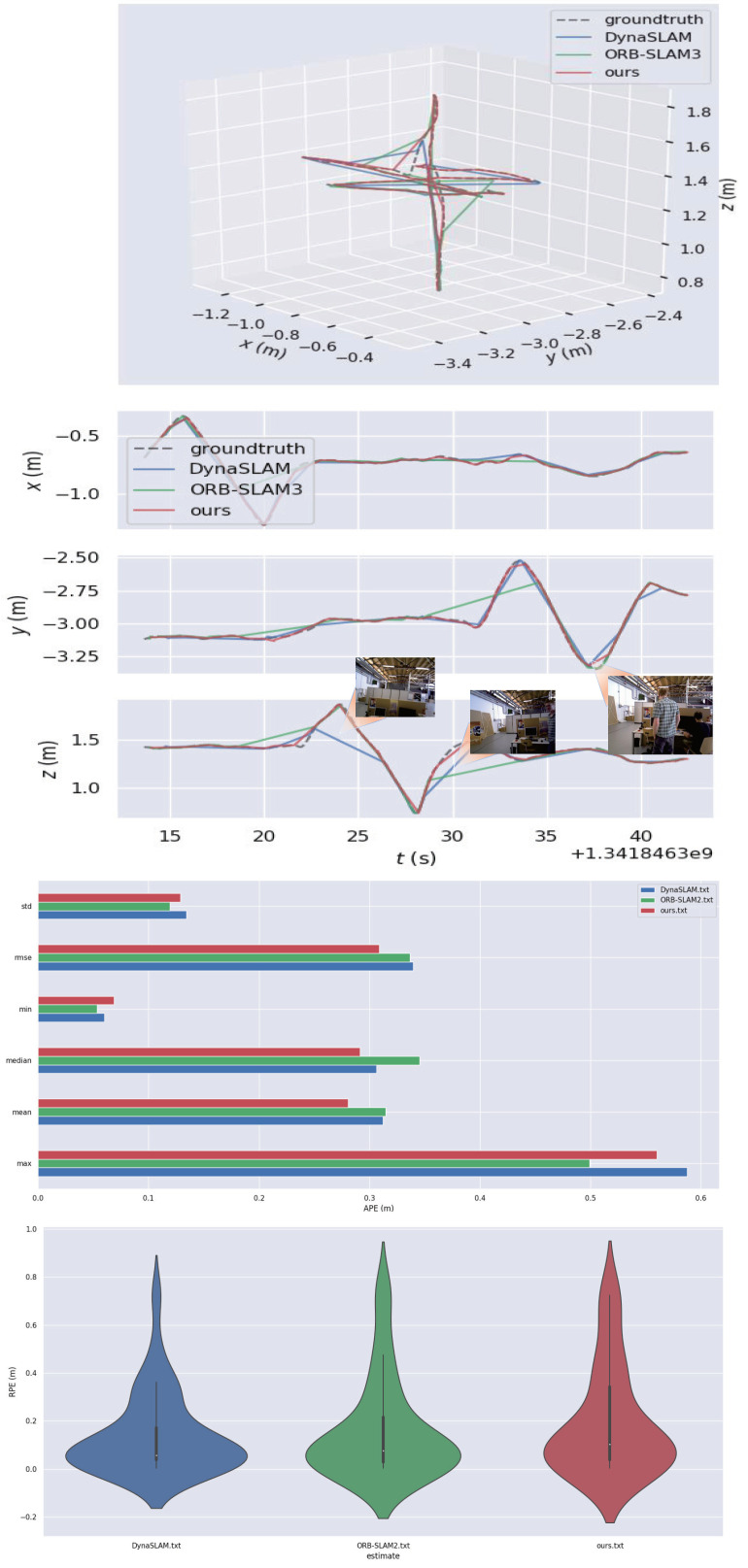
Test effect on the “rgbd_dataset_freiburg3_walking_xyz” sequence. From top to bottom, the figure shows a trajectory comparison chart, an XYZ-axis-trajectory comparison chart, the APE parameters, and an RPE data comparison chart. In highly dynamic environments, our method could still maintain high trajectory accuracy, especially in the direction of the *Y* and *Z* axes. Our method also had more advantages than the other two methods. In the bottom panel, the highest point represents the maximum RPE value while the lowest point represents the minimum RPE value. The wider the shape in the panel, the more concentrated the data. For example, in the blue shape, the RPE values are mostly clustered between 0 and 0.2.

In addition, the median and mean tracking times are also listed in Table 4; we bolded the test data obtained by the algorithm in this paper to make it easier for the reader to compare. Similar to the TUM dataset, the running speed of the proposed method was much lower than that of the ORB-SLAM2 algorithm; however, compared to the DynaSLAM algorithm, the proposed algorithm had obvious speed advantages.

### 4.3. Real Environment Test

To test the effectiveness of the proposed algorithm in indoor dynamic environments, a natural indoor environment was also tested in this study. As with DynaSLAM, natural indoor environments were recorded as simple monocular datasets for testing because they could not be run in real time. As shown in Figure 12, in indoor dynamic environments, the testing effect of behaviors ➀ and ➁, i.e., walking and sitting, caused interference in the ORB-SLAM2 system. In the feature-point extraction process, ORB-SLAM2 inevitably extracted features from the people, which caused interference in the subsequent process. In this study, we fine-tuned a Mask R-CNN, which was well-adapted to indoor environments and showed a good impact on the segmentation of prior highly dynamic objects. In addition, our method avoided extracting features from highly dynamic objects, which enabled our feature-extraction-based visual SLAM algorithm to reduce dynamic-object interference.

**Figure 10 micromachines-13-02006-f010:**
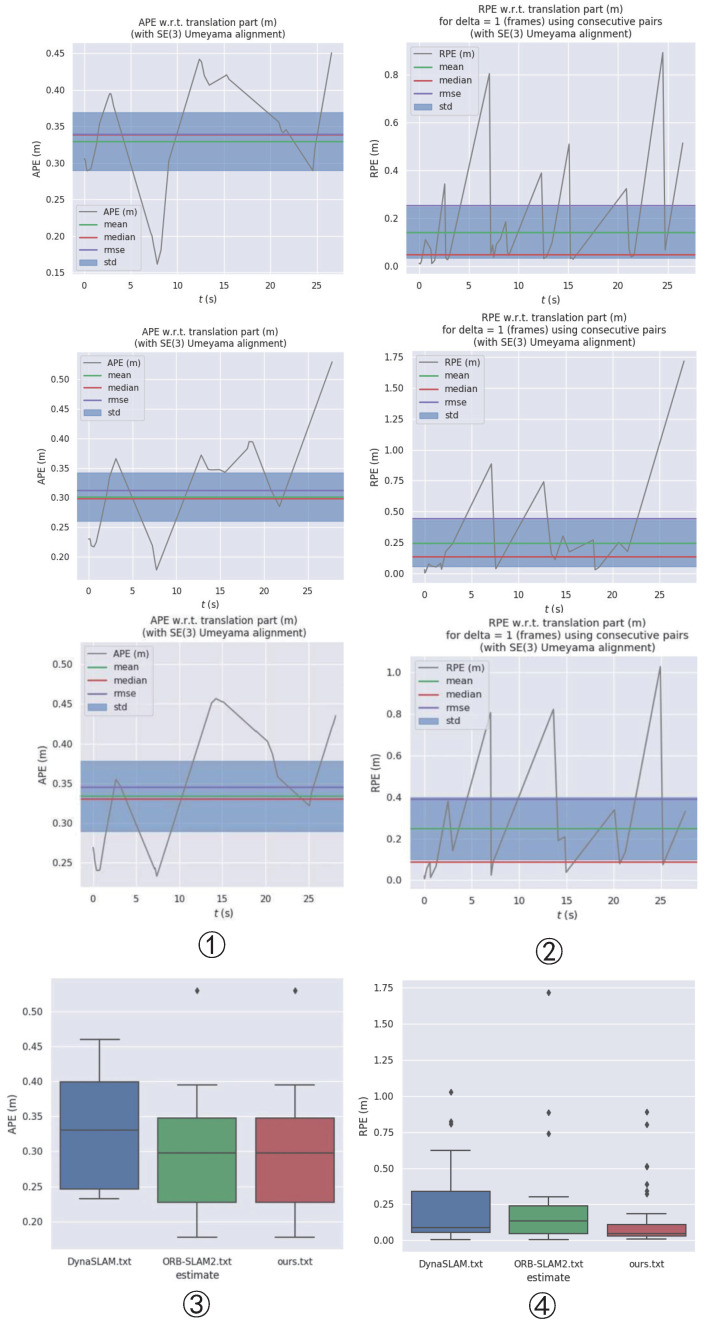
The experimental test data from the Bonn RGB-D Dynamic dataset: ➀ the APE data from the proposed, ORB-SLAM2, and DynaSLAM algorithms (from top to bottom); ➁ the RPE data from the proposed, ORB-SLAM2, and DynaSLAM algorithms (from top to bottom); ➂ in terms of APE, it can be seen that the StD, Min., and Max. values from our algorithm were significantly better than those from ORB-SLAM2, although the results for the other categories were very close; ➃ the RPE data graph, in which the highest point of the chart represents the maximum RPE value while the lowest point represents the minimum RPE value and the smaller the rectangle, the more concentrated the data (for example, in the red rectangle, the RPE values are mostly clustered between 0 and 0.1).

**Figure 11 micromachines-13-02006-f011:**
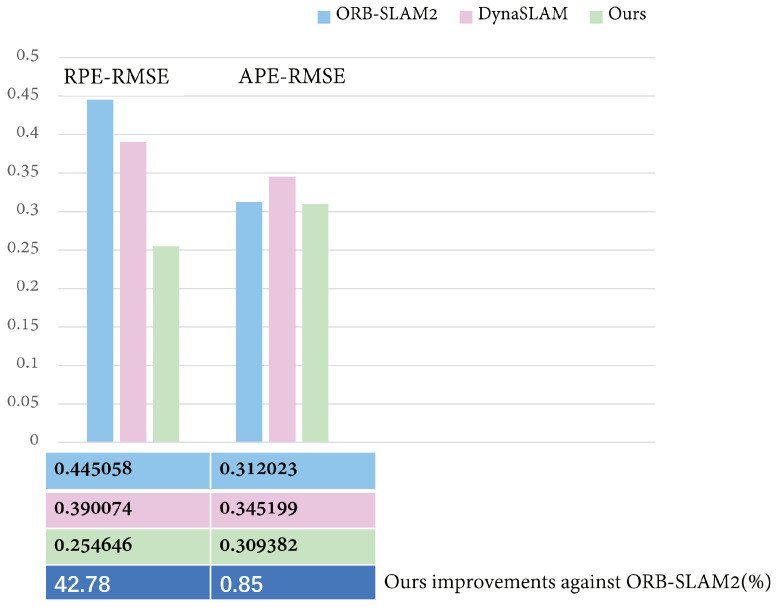
The error comparison on the “rgbd_bonn_moving_nonobstructing_box2” sequence. The specific RPE RMSE and APE RMSE values are given in the figure and the specific percentage improvements by the proposed algorithm and ORB-SLAM2 algorithm are also compared.

**Figure 12 micromachines-13-02006-f012:**
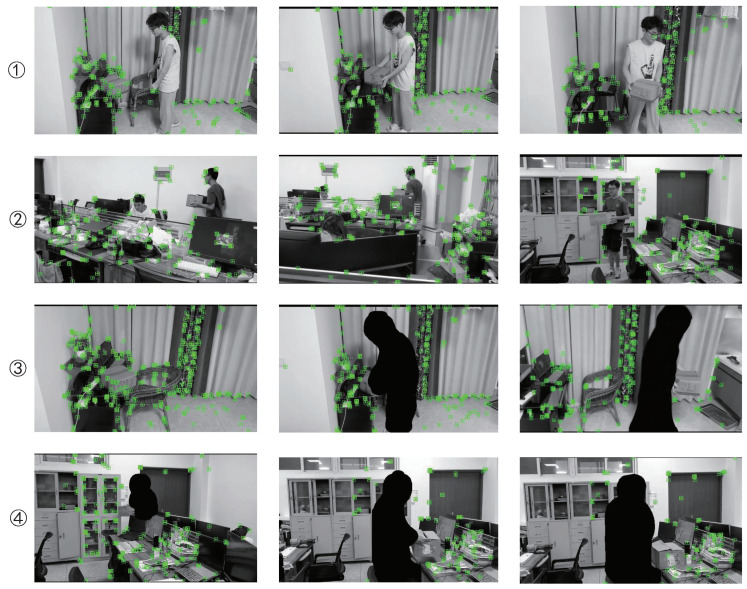
Test effect on the real indoor dynamic environment. The pictures in ➀ and ➁ lines are test renderings of ORB-SLAM2. The pictures in ➂ and ➃ lines are test renderings of our method.

## 5. Discussion and Conclusions

### 5.1. Discussion

The proposed, ORB-SLAM2, and DynaSLAM algorithms were all tested and the experimental results are shown in Table 1, Table 2, Table 3 and Table 4. As shown in Table 1, on the TUM low-dynamic dataset, the proposed method achieved lower absolute pose error (APE) and relative pose error (RPE) values compared to the classical ORB-SLAM2 visual SLAM algorithm. In particular, the proposed method was better than ORB-SLAM2 in reducing relative pose error. In addition, the proposed method was more efficient than DynaSLAM. In general, data-association errors caused by the inaccurate segmentation of dynamic features reduce the accuracy and robustness of camera pose estimation. For example, the MR-SLAM algorithm can only segment one kind of moving object in the case of multiple dynamic objects, so it is usually impossible to obtain accurate estimation results. Due to the relatively complex motion of cameras and dynamic objects, the OPF-SLAM algorithm cannot detect dynamic features effectively based on optical flow; therefore, it cannot achieve a satisfactory performance. In this study, for the low-dynamic sequences that had little impact on camera pose estimation, ORB-SLAM2 still maintained quite high accuracy and robustness. In low-dynamic sequences, the proposed method slightly improved the absolute pose error compared to ORB-SLAM2, although this effect was more evident in enhancing the relative pose error. We believe that this was because there were only a few local low-dynamic features in the data series, so they had little impact on global pose estimation. This was why ORB-SLAM2 still maintained good robustness and high accuracy.

In the highly dynamic scenarios, such as the “rgbd_dataset_freiburg3_walking_xyz” data sequence in which the environment is disturbed by people walking back and forth, the performance of ORB-SLAM2 was significantly degraded. In this case, DynaSLAM used prior semantic information to determine dynamic features and its performance was better than that of ORB-SLAM2 on this sequence. It can be seen from the RPE data that the proposed method was better than ORB-SLAM2. Still, the APE data showed that our approach did not achieve the desired effect in terms of improving local pose estimation. This could have been caused by the Mask R-CNN segmentation network focusing on pursuing a more subtle segmentation effect and that fact that pedestrians do not linger in the camera field of view for a long time.

According to the data in Table 4, the scenes in the “rgbd_bonn_moving_nonobstructin-g_box2” sequence are mainly composed of a person moving a box. The box has many feature points, so the small number of features of the person introduced little interference into the system. Our analysis showed that the proposed method could effectively improve the global trajectory accuracy and local pose accuracy in highly dynamic scenes.

### 5.2. Conclusions

In this paper, a complete dynamic monocular-visual SLAM framework was presented, which used prior semantic information to eliminate the influence of dynamic objects and the most straightforward optical sensor to improve the performance of visual SLAM systems in dynamic environments. The research offers three main contributions. Firstly, we improved a Mask R-CNN to solve the problem of existing Mask R-CNN-based semantic-segmentation frameworks not adapting well to indoor dynamic environments, which provided a good foundation for the subsequent modules. Secondly, an efficient dynamic-object detection and removal strategy was proposed which could eliminate dynamic objects simply and effectively, thereby improving system robustness and accuracy. Finally, a complete feature-based dynamic SLAM system was constructed. In our tests on the TUM dynamic indoor environment dataset, our approach performed comparably to ORB-SLAM2 in low-dynamic environments; however, in highly dynamic environments, our method significantly improved the global trajectory accuracy of SLAM systems (about 10% higher than the current state-of-the-art ORB-SLAM2 system) and outperformed the DynaSLAM framework. Our method successfully located and constructed more accurate environment maps when tested on the Bonn RGB-D Dynamic dataset and significantly outperformed both ORB-SLAM2 and DynaSLAM. The experimental results showed that the proposed method had the advantages of reliability, accuracy, and robustness in dynamic environments. It could also improve local pose estimation.

In conclusion, we explored monocular SLAM technology in real dynamic environments and successfully demonstrated the broad prospects of AI integration based on deep learning and SLAM systems. The operational efficiency of the proposed method could limit its application scenarios to a certain extent. In the future, a more lightweight semantic-segmentation network could be used instead of the Mask R-CNN or a semantic-segmentation module could be added to the server using a better hardware configuration. Additionally, real-time performance could be achieved via cloud-terminal cooperation within SLAM systems.

## Figures and Tables

**Figure 1 micromachines-13-02006-f001:**
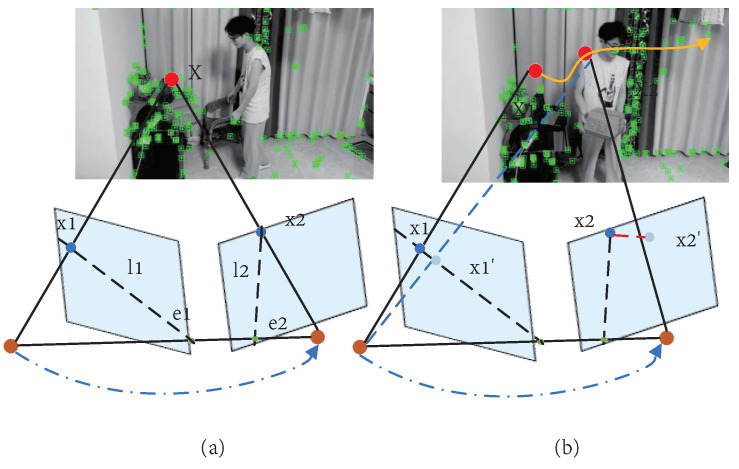
Traditional methods use geometric constraints to determine whether an object is moving: (**a**) X is a fixed point in space, so the spatial transformation relationship can be successfully obtained; (**b**) when space point X1 moves to X2, it introduces systematic errors.

**Figure 2 micromachines-13-02006-f002:**
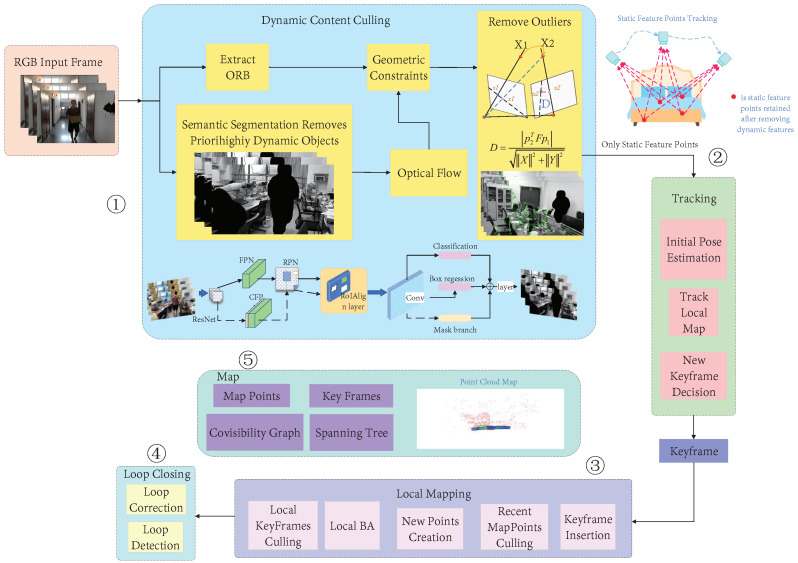
The overall framework of our approach. It can be seen that the proposed algorithm was an improved version of ORB-SLAM2 as ➁➂➃➄ is the original thread of ORB-SLAM2. Our main work is shown in ➀, i.e., the dynamic-object processing thread which we added based on ORB-SLAM2. Its function was to remove the dynamic feature points in an environment and only use the static features retained after processing for pose tracking. To reduce the influence of dynamic objects on localization and mapping accuracy in dynamic scenes, we combined the semantic segmentation method for sequential images and an optical-flow dynamic-detection module. We designed an additional segmentation thread and the optical-flow dynamic-detection module and then integrated them into our SLAM system. Our method could efficiently deal with the impact of dynamic objects on the system while retaining the superiority of ORB-SLAM2.

**Figure 3 micromachines-13-02006-f003:**
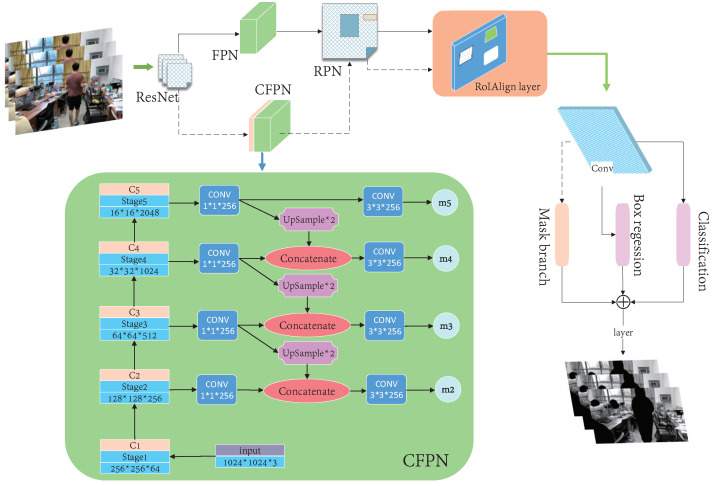
A flowchart of our improved Mask R-CNN algorithm.

**Figure 4 micromachines-13-02006-f004:**
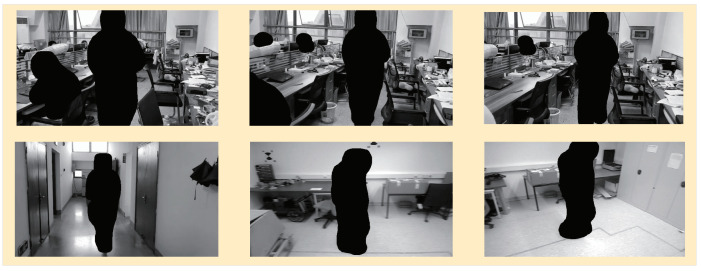
The segmentation effects of a highly dynamic semantic object. It can be seen that the improved Mask R-CNN could satisfy the semantic segmentation of a prior highly dynamic object in various dynamic environments.

**Figure 5 micromachines-13-02006-f005:**
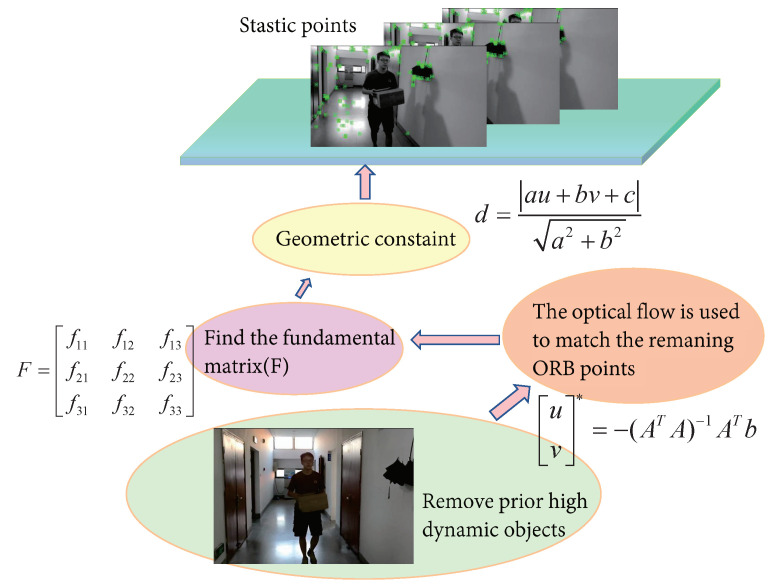
Our dynamic-object processing strategy. Since the consistency of optical flow was used to detect moving object, the selection of the optical-flow threshold greatly influenced the acquisition of dynamic point information. Therefore, the dynamic-point detection algorithm only used optical flow to track feature points and used polar constraints to determine dynamic points after obtaining specific corresponding feature points. Compared to ORB-SLAM2, our system had a more stringent feature-point selection strategy, which ensured more correct matching points, thereby meeting the requirements of reducing trajectory errors and achieving the stable operation of the system in dynamic scenarios.

**Figure 6 micromachines-13-02006-f006:**
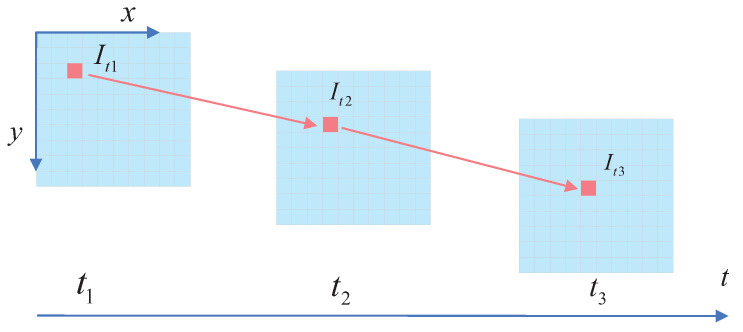
A schematic diagram of optical flow. The optical-flow method is used to avoid the frequent calculation of feature points. In practice, the function of LK optical flow is to track feature points. Compared to extracting feature points for each frame, LK optical flow only needs to extract feature points once and only the subsequent video frames need to be tracked, which saves a lot of time.

**Figure 7 micromachines-13-02006-f007:**
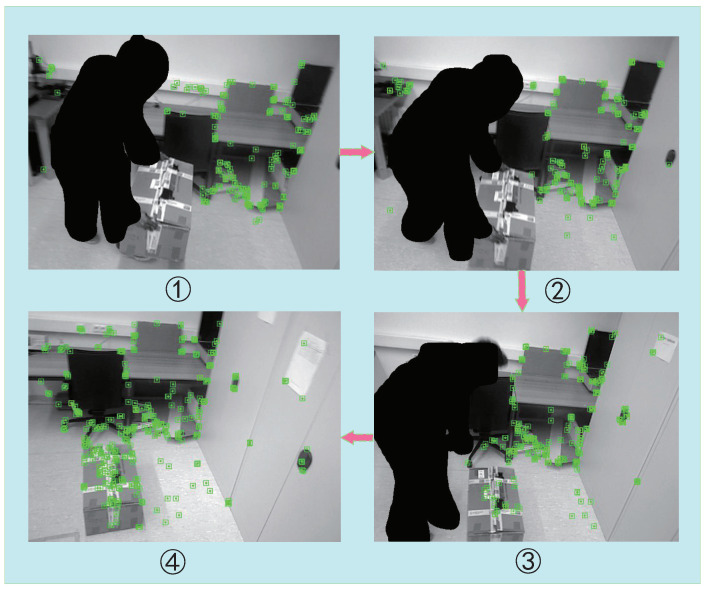
A diagram of our moving-object processing strategy. From ➀ to ➃, as a complete process, we can see that the human, as a prior highly dynamic object, was segmented and no features were proposed in the process. In ➀, the box (an object that can be moved under human influence) was held and carried by people, so the package was in motion. Our dynamic-object processing strategy was adopted to avoid the box’s proposed feature effectively. In ➂, the human put the box on the ground, so the package changed from dynamic to static. Therefore, according to our method, the static features were extracted from the box again. In ➃, after the person ultimately left, the whole environment became static and the system continued to operate stably.

**Figure 8 micromachines-13-02006-f008:**
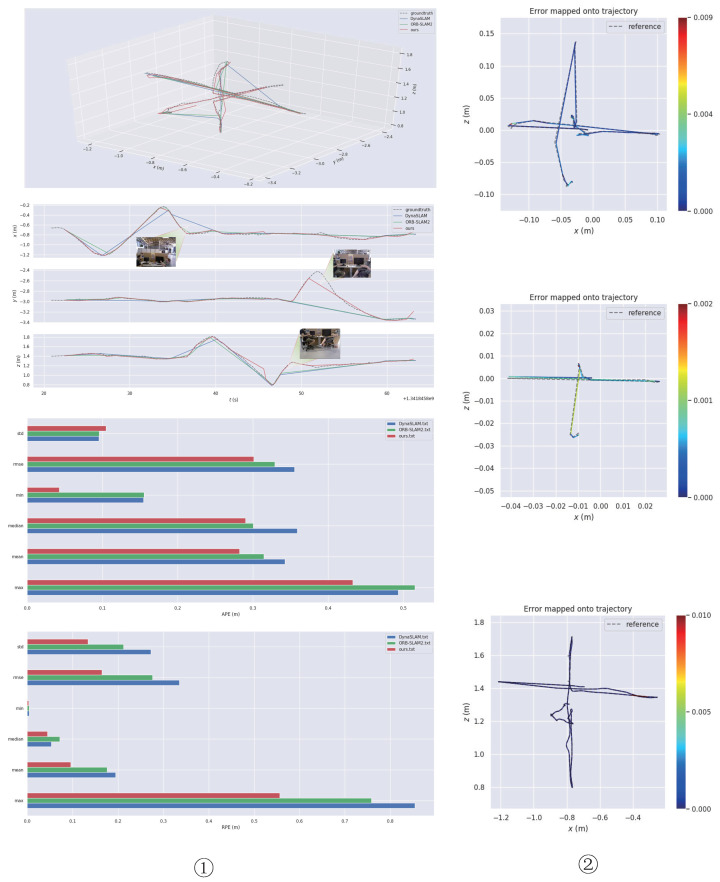
Test effect on the “rgbd_dataset_freiburg3_sitting_xyz” sequence. From top to bottom, the ➀ column shows a trajectory comparison chart of the three algorithms, an XYZ-axis trajectory comparison chart, an APE-data comparison chart, and an RPE-angle comparison chart. In terms of the APE data, it can be seen that there was little difference in performance between the three algorithms in low-dynamic scenarios, although our method had a slight advantage. In contrast, our method was more effective in reducing RPE. From top to bottom, the ➁ column shows trajectory-error plots for ORB-SLAM2, DynaSLAM, and our method.

**Table 1 micromachines-13-02006-t001:** The specific APE and RPE data for the “rgbd_dataset_freiburg3_sitting_xyz” sequence.

Method	APE (m)				RPE (m)			
	Mean	Median	RMSE	StD	Mean	Median	RMSE	StD
ORB-SLAM2	0.3148	0.3004	**0.329**	0.096	0.1761	0.0713	**0.2757**	0.2122
DynaSLAM	0.3425	0.3589	**0.3555**	0.0953	0.1948	0.0531	**0.3348**	0.2723
Ours	0.2785	0.2869	**0.2973**	0.1041	0.1011	0.0506	**0.1679**	0.1341

**Table 2 micromachines-13-02006-t002:** The specific APE and RPE data for the “rgbd_dataset_freiburg3_walking_xyz” sequence.

Method	APE (m)				RPE (m)			
	Mean	Median	RMSE	StD	Mean	Median	RMSE	StD
ORB-SLAM2	0.3147	0.3452	**0.3375**	0.1191	0.1589	0.0753	**0.2572**	0.2023
DynaSLAM	0.3119	0.3065	**0.3396**	0.1341	0.1403	0.0577	**0.2162**	0.1644
Ours	0.2805	0.2913	**0.3087**	0.1288	0.1017	0.1022	**0.2911**	0.2909

**Table 3 micromachines-13-02006-t003:** The tracking time performance of the three algorithms.

Tracking Time (s)	TUM Sequence					
	sitting_xyz			walking_xyz		
	ORB-SLAM2	Ours	DynaSLAM	ORB-SLAM2	Ours	DynaSLAM
Median	0.01843	**4.23753**	4.54982	0.01861	**3.99734**	4.23104
Mean	0.01942	**4.20628**	4.51254	0.02148	**4.01317**	4.30717

**Table 4 micromachines-13-02006-t004:** The specific APE and RPE data for the Bonn dataset.

rgbd_bonn_moving_ nonobstructing_box2		Ours	ORB-SLAM2	DynaSLAM
RPE (m)	Mean	0.141	0.243	0.249
	Median	0.048	0.135	0.088
	RMSE	**0.254**	**0.445**	**0.39**
	SSD	2.464	4.753	3.195
	StD	0.212	0.372	0.3
APE (m)	Mean	0.33	0.301	0.334
	Median	0.338	0.298	0.33
	RMSE	**0.309**	**0.312**	**0.345**
	SSD	4.492	2.439	2.622
	StD	0.08	0.082	0.088
Median Tracking Time (s)		**4.20007**	0.01697	4.42606
Mean Tracking Time (s)		**4.19947**	0.01764	4.47998

## Data Availability

The data is temporarily unavailable.

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
