# Peer review of "A Monocular-Visual SLAM System with Semantic and Optical-Flow Fusion for Indoor Dynamic Environments"

_micromachines, 2022, doi:10.3390/mi13112006_

Round 1
Reviewer 1 Report
This paper designed a monocular visual SlAM for dynamic indoor environments. In order to reduce the influence of dynamic objects in feature tracking, the authors applied a fine turn Mask R-CNN to remove highly dynamic objects in images. The rest of the features extracted from dynamic objects are filtered by using the optical flow method with distance checking for the corresponding feature between images. The proposed algorithm was evaluated using TUM and Bonn datasets. The paper is well-written and easy to follow.
However, the results shown in this manuscript are questionable.
1. In Figure 10, the XYZ axis trajectory comparison chart shows some problems. The trajectories from ground truth, DynaSLAM, ORB-SLAM2, and ours do not start at the same position. In this case, the length of running using each algorithm is different. Therefore, the errors of the mean and median for each algorithm cannot be compared.
2. Similar to question 1, in Figure 11, subfigures 1 and 2, the time running for each algorithm is different.
3. In Figure 13, the last two images in row 4 are the same.
Author Response
We are grateful to the anonymous reviewer for his\her kind help and constructive comments which helped improving the presentation of the paper.
We value this opportunity to receive reviewer guidance.
We look forward to hearing from you regarding our submission. We would be glad to respond to any further questions and comments that you may have.
Please check the attachment for the specific modification.Thanks again for your kindly help.
Thank you!
All authors

Reviewer 2 Report
This paper proposes robust SLAM system using semantic information and geometric methods in dynamic environments. Overall, I think the contribution of this paper is sufficient for publication, and the next part needs to be checked.
1. It is necessary to change the title of the manuscript to be more specific. It is difficult to guess which methodology is being used just by looking at the title.
2. The research motivation and background were well explained in the introduction section.
3. Relevant research is well organized up to the latest research.
4. It is necessary to check whether Equations (2) and (3) are properly defined. According to the current definition, is e_k,j equals to v_k,j? Also, isn't it necessary to distinguish between camera poses and landmark poses? If it is a widely used formula, please mention an appropriate reference.
5.
Minor corrections recommendation
- It is more common to write the first abbreviation in parentheses. Currently, the description rather than the abbreviation is enclosed in parentheses.
- Make sure to leave a space before citation in the text.
- Figure 5 has a typo and I don't think it's a necessary figure.
- There is a detailed explanation of optical flow in 3.3.1, but it would be better to simply cite the related paper and omit it.
Author Response
We are grateful to the anonymous reviewer for his\her kind help and constructive comments which helped improving the presentation of the paper.
We value this opportunity to receive reviewer guidance.
On this basis, we re-polished the article in English to make our article more professional. We look forward to hearing from you regarding our submission. We would be glad to respond to any further questions and comments that you may have.
Please check the attachment for the specific modification. Thanks again for your kindly help.
Thank you!
All authors

Round 2
Reviewer 1 Report
Some minor comments,
1. In Table 4, the authors need to explain why the APE of ORB-SLAM2 shows better than the presented algorithm.
2. What is the rule that the authors based on to bold numbers in tables? Maximum number? Need to clarify which numbers in each table need to be bolded.
Author Response
我们感谢匿名审稿人的善意帮助和建设性意见,这有助于改进论文的呈现方式。
我们珍惜有机会接受审稿人指导。
我们期待收到您关于我们提交的意见。我们很乐意回答您可能提出的任何进一步问题和意见。
再次感谢您的帮助!
所有作者
